# Ecological and geographical overlap drive plumage evolution and mimicry in woodpeckers

Eliot T. Miller[1], Gavin M. Leighton [1,2], Benjamin G. Freeman[3], Alexander C. Lees [1,4] & Russell A. Ligon [1]

Organismal appearances are shaped by selection from both biotic and abiotic drivers. For example, Gloger's rule describes the pervasive pattern that more pigmented populations are found in more humid areas. However, species may also converge on nearly identical colours and patterns in sympatry, often to avoid predation by mimicking noxious species. Here we leverage a massive global citizen-science database to determine how biotic and abiotic factors act in concert to shape plumage in the world's 230 species of woodpeckers. We find that habitat and climate profoundly influence woodpecker plumage, and we recover support for the generality of Gloger's rule. However, many species exhibit remarkable convergence explained neither by these factors nor by shared ancestry. Instead, this convergence is associated with geographic overlap between species, suggesting occasional strong selection for interspecific mimicry.

[1] Cornell Lab of Ornithology, 159 Sapsucker Woods Rd., Ithaca, NY 14850, USA. [2] Department of Biology, SUNY Buffalo State College, Buffalo, NY 14213, USA. [3] Department of Zoology, University of British Columbia, #4200-6270 University Blvd, Vancouver, BC V6T1Z4, Canada. [4] School of Science and the Environment, Manchester Metropolitan University, Manchester M1 5GD, UK. Correspondence and requests for materials should be addressed to E.T.M. (email: etm45@cornell.edu)

The coloration and patterning of organisms is shaped over evolutionary timescales by a variety of factors, both biotic and abiotic, including temperature and humidity[1–5]. Gloger's rule, for example, describes the prominent ecological pattern wherein more pigmented populations are found in more humid areas[1,6–8]. Sexual selection can push organisms to become conspicuous, whilst the risk of predation can select for inconspicuous visual signals[9–11].

The external appearances of animals are subject to frequent study because such work has the power to shape our understanding of phenotypic evolution. Yet, our understanding of how factors such as climate and biotic interactions with predators, competitors, and mates combine to influence evolutionary outcomes across large radiations remains rudimentary. This is true even for birds, regular subjects of research on phenotypic evolution[12,13]. Here, we employ a phylogenetic comparative framework, coupled with remote-sensing data and a large citizen science dataset, to examine the combined effects of climate, habitat, evolutionary history, and community composition on plumage pattern and colour evolution in woodpeckers (Picidae). This diverse avian clade of 230 bird species is an excellent group in which to examine the evolution of external appearances because they occupy a broad range of climates across many

habitats. Woodpeckers also display a wide range of plumages, from species with boldly pied patterns to others with large bright red patches, to still others that are entirely dull olive (Fig. 1). Furthermore, woodpeckers exhibit several cases of ostensible plumage mimicry[14,15], highlighted by a recent time-calibrated phylogeny[16]. Although qualitatively compelling, it is unclear if these events can be explained simply as consequences of shared climate, habitat, and evolutionary history. Regardless of the answer to this question, these purported mimicry events and the impressive variation in plumage among woodpecker species provide the raw variation that we examine here to disentangle the contribution of the various abiotic and biotic factors that drive plumage evolution.

We find that climate and habitat exert strong influences on woodpecker plumage. Species from humid areas, for example, tend to be darker and less boldly patterned than those from drier regions, and thus offer compelling support of Gloger's rule. These factors and shared evolutionary history explain some of the variation in woodpecker plumage, but they are insufficient to explain some of the dramatic convergence seen between various sympatric woodpecker species. Instead, sympatry in and of itself appears to drive certain species pairs to converge in plumage,

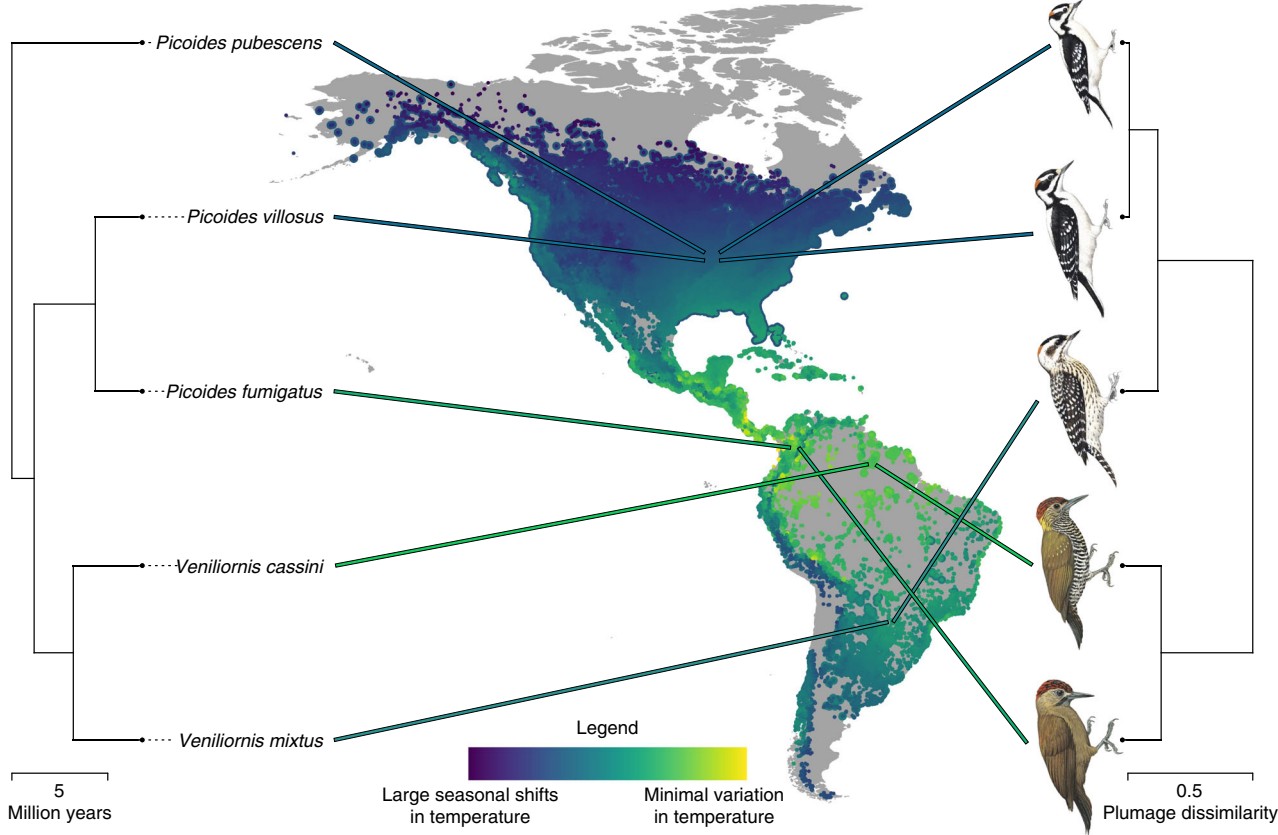

**Fig. 1** Evolutionary relationships and plumage similarity among exemplar species. Climate partially determines variation in woodpecker plumage. Lines lead from tips of phylogeny (left) to centroid of each species' geographic distribution and are coloured according to mean climate regime of each species. These species shared a common ancestor ~ 6.5 mya. The colour scale depicts a gradient from warm (yellow) to seasonally cold regions (blue). eBird records for these species are plotted in the same colours as large points on the map. All other eBird woodpecker records are overlaid as smaller points and coloured similarly. Plumage dendrogram (right) shows the plumage dissimilarity relationships among the same set of species. *Veniliornis mixtus*, long classified as a member of *Picoides*, is inferred to have invaded seasonal climates in the southern hemisphere, and accordingly evolved bold black and white plumage. *Picoides fumigatus*, long classified as a member of *Veniliornis*, is inferred to have invaded warm climates near the equator, and accordingly evolved dark, subtly marked plumage. *Picoides pubescens* and *P. villosus* are rather distantly related but largely sympatric; they are inferred to have converged on one another in plumage above and beyond what would be expected based on shared climate, habitat, and evolutionary history. Traditional scientific names are used in this figure to aid explanation, but the illustrated species are currently all members of an expanded clade, *Dryobates*. Illustrations © HBW Alive/Lynx Edicions, map by authors

lending credence to the notion that these species are true avian plumage mimics.

## Results

**Multidimensional, distance-based approaches**. To investigate how climate, habitat, social interactions, and evolutionary history determine woodpecker plumage outcomes, we used multidimensional-colour and pattern-quantification tools to measure species' colouration and patterns, quantifying species' plumages from a standardized source (Figs. 2 and 3)[12,17,18]. Evidence suggests that pattern and colour are likely processed separately in vertebrate brains, with achromatic (i.e., luminance) channels used to process pattern information[19], and differential stimulation of cones used to encode chromatic information[20]. While both plumage colour and pattern are inherently multivariate, we reduced this complexity into a composite matrix of pairwise species differences to address whether purported convergences were a mere by-product of shared evolutionary history or, if not, whether shared climate, habitat, or geographic overlap could explain these events. We incorporated the potential for interactions between pairs of species into the analysis by quantifying pairwise geographic range overlap using millions of globally crowd-sourced citizen science observations from eBird;[21] species in complete allopatry have no chance of interacting, while increasing degrees of sympatry should correlate with the probability of evolutionarily meaningful interactions.

Variation in climate (multiple distance matrix regression, $r = 0.055$, $p = 0.006$), habitat ($r = 0.106$, $p = 0.007$) and, to a lesser degree, phylogenetic relationships ($r = 0.001$, $p = 0.015$) were all correlated with woodpecker plumage similarity scores. These results were robust to phylogenetic uncertainty (Supplementary Fig. 1). In short, woodpecker species in similar climates and habitats tend to look alike, even after accounting for shared ancestry. However, beyond the influences of habitat, climate, and evolutionary relatedness, we also found that close sympatry was a strong predictor of plumage similarity for the most similar-looking species pairs (Fig. 4). We interpret this result as evidence for multiple instances of plumage mimicry per se, transcending broader patterns of plumage convergence driven by similar environmental conditions. Following this result, we developed a method (see 'Identification of putative plumage mimics' in the Methods) to identify the species pairs that powered this result. Using this method, we validated many previously qualitatively identified mimicry complexes, including the Downy-Hairy Woodpecker system (Fig. 1)[22], repeated convergences between members of *Veniliornis* and *Piculus*[23], *Dinopium* and *Chrysocolaptes*, *Dryocopus* and *Campephilus*[24], and the remarkable convergence of *Celeus galeatus* on *Dryocopus* and *Campephilus*[15]. Collectively, these distance matrix-based analyses provide a powerful tool to identify and understand the various factors that drive evolutionary patterns of convergence and divergence.

These previous analyses focused on the whole-body phenotype, however it is possible that environmental and social drivers of plumage operate in unique ways on different plumage patches[12]. To investigate this possibility, we ran additional analyses for each of three different body segments: (1) the back, wings, and tail; (2) the head; and (3) the breast, belly and vent.

The whole-body results were largely recapitulated by these body-region-specific results, with subtle but notable differences. In particular, range overlap was particularly strongly associated with driving convergence in back plumage similarity, while genetic and climate similarity were not implicated, and genetic similarity was particularly closely associated with belly and head plumage similarity, while habitat (belly and head) and climate (belly) were not involved. To gain further insight into the evolutionary drivers of particular colours and patterns, we subsequently employed species-level phylogenetic comparative approaches.

**Species-level phylogenetic comparative approaches**. Considering the full-body plumage phenotype, we found that precipitation drives global patterns of pigmentation and patterning in woodpeckers. In particular, darker species tend to inhabit areas of higher annual precipitation (phylogenetic generalized least squares [PGLS] $r^2 = 0.084$, $p < 0.001$, Fig. 5a), supporting Gloger's rule[1,8]. In addition, high precipitation is also associated with reduced patterning (PGLS $r^2 = 0.170$, $p < 0.001$, Fig. 5c), augmenting the generality of Gloger's rule. While this pattern of dark populations occurring in areas of high precipitation is so well known as to be considered a "rule", few large-scale comparative studies have quantitatively assessed this across a large radiation[8]. The mechanism underlying Gloger's rule remains debated, but proposed drivers include improved background matching[25] in response to increased predation pressure in humid environments[26], and defence against feather-degrading parasites[27]. There are some boldly marked woodpecker species in humid areas, but they invariably achieve these conspicuous phenotypes with minimal use of white plumage. This hints at the existence of an evolutionary trade-off wherein Gloger's rule is due to the ability of melanin to forestall feather wear (e.g., by inhibiting parasites prevalent in humid areas[27]), which subsequently narrows the breadth of means by which humid forest-inhabiting woodpeckers can achieve bold plumage phentoypes. Alternatively, unconcealed large white plumage patches might simply subject humid forest-dwelling birds to evolutionarily unacceptable levels of predation (the abundance and preferences of predators such as *Accipiter* hawks would shed more light on this issue, given that increasing body mass is associated with increasingly bold plumage patches in woodpecker, Fig. 5c). While additional research is necessary to delineate the mechanism(s) responsible, our results expand the generality of Gloger's rule and

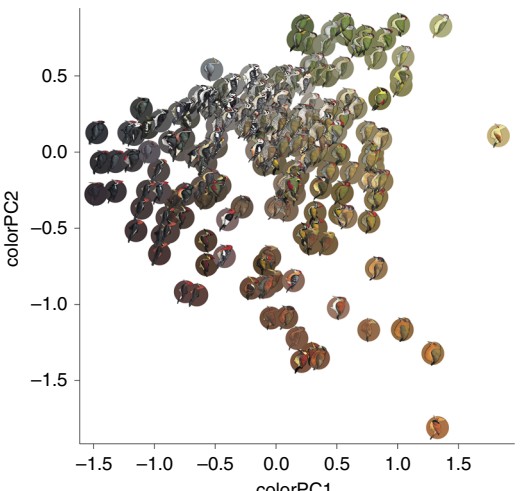

**Fig. 2** Principal components analysis (PCA) of species-averaged woodpecker colour values. Principal component one (colourPC1) explains 45% of the variation in measured colour scores. Higher PC1 scores correspond to greater luminance values, and more yellow and less blue. Principal component two (colourPC2) explains an additional 36% of variation in overall colour scores. Higher PC2 scores correspond to more green and less red colouration. Coloured circles behind each woodpecker species correspond to the average CIE L*a*b scores for the 1000 randomly drawn colour samples from that species. These PCA values are used for species-level analyses (e.g., Fig. 5). Illustrations © HBW Alive/Lynx Edicions

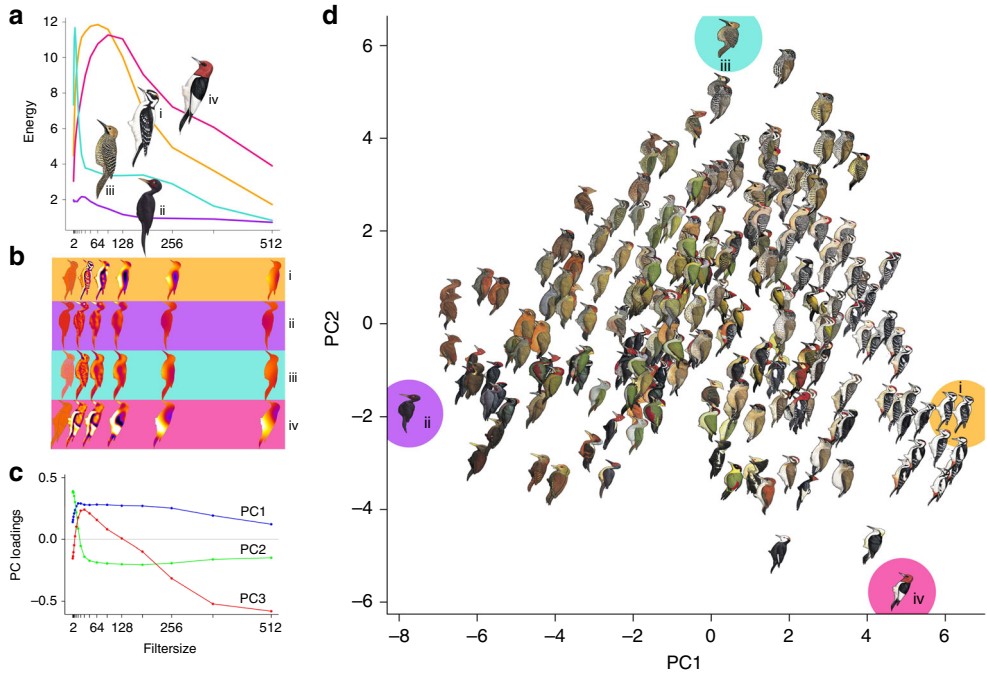

**Fig. 3** Major axes of plumage pattern variation, summarized with principal components analysis (PCA). From granularity analysis, these values are used for species-level approaches. **a** Pattern energy spectra for exemplar species characterized by maximally divergent PC scores (i = *Dryobates villosus*, ii = *Mulleripicus funebris*, iii = *Colaptes fernandinae*, iv = *Melanerpes erythrocephalus*). These show information on relative contributions of different granularities to overall appearance. **b** Pattern maps of exemplar species enable visualization of energy at subsets of isotropic band-pass filter sizes (2, 32, 64, 128, 256, and 512 pixels). **c** Principal component loadings of pattern energy spectra across filter size for all species reveals how pattern elements of different sizes influence PC scores. **d** Pattern PC1 and PC2, collectively, account for 82.1% of variation across woodpeckers. Species (i–iv) illustrate the extremes of variation along PC1 and PC2: (i) exhibits high energy scores across most pattern element sizes, with small, medium, and large pattern elements; (ii) has low energy scores across the spectrum, with few pattern elements of any size; (iii) has many small pattern elements and few of any other sizes; (iv) has only medium and large size pattern elements. Illustrations © HBW Alive/Lynx Edicions

show that it may be involved in phenotypic convergence among disparate lineages inhabiting similar forests.

Seasonality, in addition to average annual precipitation and temperature, also exerts significant influence on woodpecker plumage. The gradient from dark- to light-plumaged woodpeckers (colorPC1) was best explained by a model that included body mass, latitude, and seasonality in precipitation. Darker birds are larger, are found at lower latitudes, and in climates that receive considerable precipitation throughout the year (PGLS $r^2 = 0.084$, $p < 0.001$, Fig. 5a). The gradient from red to green plumaged woodpeckers (colorPC2) was best explained by a model that included variation in temperature seasonality, and that included the dichotomy between open habitats and closed forests. Specifically, green birds tend to be found in climates that experience seasonal temperature fluctuations, and in open habitats (PGLS $r^2 = 0.073$, $p < 0.001$, Fig. 5b). Seasonality also drives woodpecker patterning, and boldly marked birds (patternPC1) tend to be found in seasonal climates, open habitats, and temperate forests (PGLS $r^2 = 0.170$, $p < 0.001$, Fig. 5c). We had suspected that variation along the gradient from species with large plumage elements to those with barring and spotting (patternPC2) might be associated with sexual selection, but after accounting for body mass, patternPC2 was not associated with sexual size dimorphism; instead, more finely marked birds tend to be smaller and found in lower reflectance habitats such as rainforests (PGLS $r^2 = 0.043$, $p = 0.025$, Fig. 5d). Like those results from the multiple distance matrix regressions, these results were robust to phylogenetic uncertainty (Supplementary Figs. 3–6).

Results were largely similar when considering the drivers of plumage variation for specific body parts, particularly for back plumage coloration and patterning (Supplementary Fig. 7). Yet these body-part-specific analyses did provide additional insights

and investigating the mechanistic bases for these relationships should prove fruitful future research grounds. For example, red-headed species tend to be found in closed habitats, whereas black-, white-, and grey-headed species tend to be found in open habitats (Supplementary Fig. 8). In dark-headed species, including those with red heads, females tended to be heavier than males, whereas species with yellow and pale heads tend to have heavier males. Additionally, red-bellied species are most often found in forested habitats, species with boldly patterned bellies tend to have males that are heavier than females, and species with bellies patterned with large plumage patches (as opposed to fine barring) tend to be heavier and live in open habitats (Supplementary Fig. 9).

## Discussion

Although climate and habitat appear responsible for some of the convergence in external appearance in woodpeckers, our analyses confirmed the decades-old suggestions[28] that some species have converged above and beyond what would be expected based only on selection pressures from the environments they inhabit. Sympatry, a proxy for the likelihood of evolutionarily meaningful interspecific interactions, was a strong predictor of plumage similarity for species exhibiting large geographic range overlaps (Fig. 4). We interpret this finding as evidence that the pattern of convergence we document is true mimicry, i.e., phenotypic evolution by one or both parties in response to a shared signal receiver[3,4]. Indeed, our study almost certainly underestimates the degree to which close sympatry leads to mimicry in woodpeckers, since some postulated mimetic dyads are well known to track one another at the subspecific level, which we could not account for

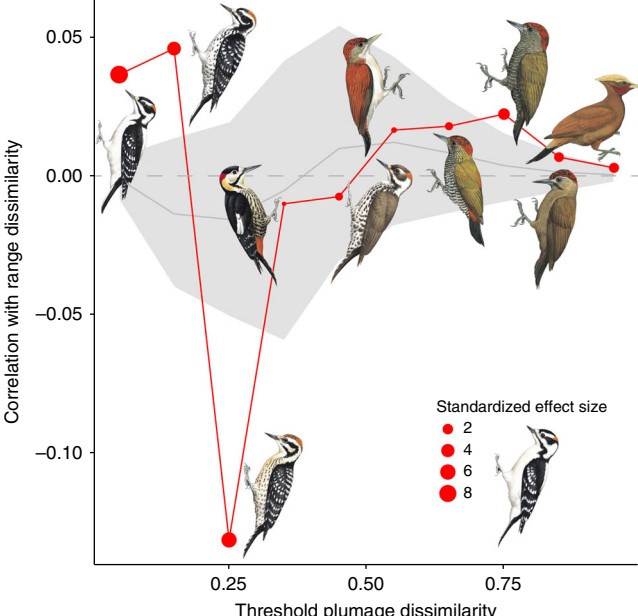

**Fig. 4** Modified Mantel correlogram shows how correlation of range and plumage changes across varying plumage dissimilarity. For dyadic comparisons with plumage dissimilarities of 0–0.2, geographic range overlap *per se* is statistically significantly associated with increasing plumage phenotype matching between already similar looking species pairs. The relationship was reversed at intermediate levels of plumage dissimilarity, where geographic range overlap is statistically significantly associated with decreasing plumage phenotype matching between somewhat similar looking species pairs (plumage dissimilarities of 20–30%). Illustrations show examples of species with plumage dissimilarities in the range indicated as compared with the Downy Woodpecker (*Dryobates pubescens*) inset in the legend. The red line shows the observed correlation coefficients, while the shaded grey area shows the expected correlation coefficients given the simulations described in the text. The size of the red circle corresponds to the standardized effect size of the observed correlation coefficient; values greater than +/−1.96 deviate beyond 95% of simulated values. Illustrations © HBW Alive/Lynx Edicions

here. Moreover, recent taxonomic revision of *Chrysocolaptes*[29] not yet matched by equivalent efforts in *Dinopium* meant that we could not completely capture the breadth of plumage matching events in this mimicry complex (e.g., the extraordinary convergence between the maroon-coloured Sri Lankan endemics *C. stricklandii* and *D. benghalense psarodes*). There are two contested questions regarding plumage mimicry: whether it truly occurs[24,28,30] and, if it does, what process(es) drive the pattern[31–34]. Here we have shown that plumage mimicry does indeed occur and is pervasive across the woodpecker evolutionary tree, indicating that the processes deserve further study.

Given the strong evidence that mimicry occurs in woodpeckers—a taxon with no known chemical defences—we predict renewed research interest in understanding the mechanisms responsible for these patterns. Only a handful of other avian studies[22,35] have empirically demonstrated that convergence—on the scale which we document among woodpeckers—is not exclusively a product of shared evolutionary history or environmental space, but this has not deterred more than a century-worth of careful rumination over the mechanisms responsible for the compelling patterns[28,32,33,36–38]. Recently, it has been shown[34] that the smaller species in plumage mimicry complexes may derive a benefit by fooling third parties into believing they are the socially dominant model species, and this remains the best empirically

supported hypothesis in birds, but experimental work is needed to adequately quantify the selective advantage mimicry might confer. Relatedly, it remains unknown how distantly related lineages achieve plumage convergence genomically. Are multiple mutations required, each of which increases the degree of plumage convergence? Or might selection act on genetic modules controlled by a few loci shared across woodpeckers? Or might rare hybridization events between sympatric species have resulted in adaptive introgression of relevant plumage control loci [23,39]?

In summary, habitat and climate are strong determinants of woodpecker plumage. Shared evolutionary history shapes plumage phenotypes, but selective factors have driven plumage divergence far beyond that expected of simple evolutionary drift. Perhaps most notably, the plumage similarity predicted by shared climate, habitat, and evolutionary history is insufficient to explain the large number of cases we detected of closely sympatric but distantly related woodpecker species converging in colour and pattern. Woodpeckers appear to be involved in globally replicated mimicry complexes similar to those in well-studied groups such as butterflies[39], and while woodpeckers are among the most conspicuous avian plumage mimics, others such as toucans exhibit qualitatively similar patterns[14]. Assessing how these evolutionary constraints and selective pressures have operated in concert is a research question that has only recently become more tractable with the advent of large, time-calibrated molecular phylogenies, massive distributional databases such as eBird, and powerful computing techniques like pattern analysis. It seems likely that different clades have been more or less influenced by factors such as climate, habitat, and social interactions, and understanding how and why these factors differ among clades should be a particularly fertile line of enquiry.

## Methods

**Taxonomic reconciliation and creation of complete phylogenies.** A time-dated phylogeny containing nearly all known woodpecker species was recently published by Shakya and colleagues[16]. As described below, we used (and verified the use of) illustrations from the Handbook of the Birds of the World (HBW) Alive[17] to quantify woodpecker plumage, and we used eBird[21], a massively crowd-sourced bird observation database, to define spatial, climate, and habitat overlap between species. Each of these references uses a slightly different taxonomy. Our goal was to use the species-level concepts from the most recent eBird/Clements taxonomy[40] as our final classification system.

To reconcile these three taxonomies (HBW, Shakya et al., and eBird/Clements), we obtained a set of 10,000 credible trees, kindly provided by Shakya[16]. We checked to ensure that each tree contained no polytomies, was ultrametric, and included the same set of tip labels as the other trees. After passing these checks, we discarded the first 30% of trees as burn-in, then sampled 1000 of the remaining trees. We extracted a list of the tip labels from the first tree, then determined to which eBird taxon this label was best applied. Across the set of 1000 credible trees we then swapped out the original tip labels for their eBird taxonomic identities. For each credible tree, we then randomly dropped all but one of any taxon represented by more than one terminal. We then worked in the opposite direction and identified all woodpecker taxa according to eBird. This process made it clear which species, as recognized by eBird, were missing from the Shakya tree.

Twenty-one such missing taxa were identified: *Picumnus fuscus, P. limae, P. fulvescens, P. granadensis, P. cinnamomeus, Dinopium everetti, Gecinulus viridis, Mulleripicus fulvus, Piculus simplex, Dryocopus hodgei, Melanerpes pulcher, Xiphidiopicus percussus, Veniliornis maculifrons, Dendrocopos analis, Dendrocopos ramsayi, Colaptes fernandinae, Chrysocolaptes festivus, C. xanthocephalus, C. strictus, C. guttacristatus,* and *C. stricklandi.* We added these using the R package *addTaxa*[41], and taxonomic hypotheses outlined in previous work (reviewed in Shakya et al.[16]). Eighteen of these taxa have fairly precise hypothesized taxonomic positions which we were able to leverage to carefully circumscribe where they were bound into the tree. As an example, *Dinopium everetti* was recently split from *D. javanense*, so it was simply added as sister to the latter species. The precise phylogenetic positions of the remaining three taxa are less well known. For these, we first added *C. fernandinae* as sister to *Colaptes sensu stricto* (as previously found[42]), then added *Piculus simplex* into the clade *Piculus + Colaptes*, as previous work showed some members of the former genus to actually belong to the latter[42]. We added *X. percussus* as sister to *Melanerpes striatus*;[16] and we added *P. cinnamomeus* into *Picumnus* while ensuring that the Old World *P. innominatus* remained sister to the rest of the genus (it is very likely the New World *Picumnus* form a clade). Each of the 1000 resulting trees contained 230 species. As described

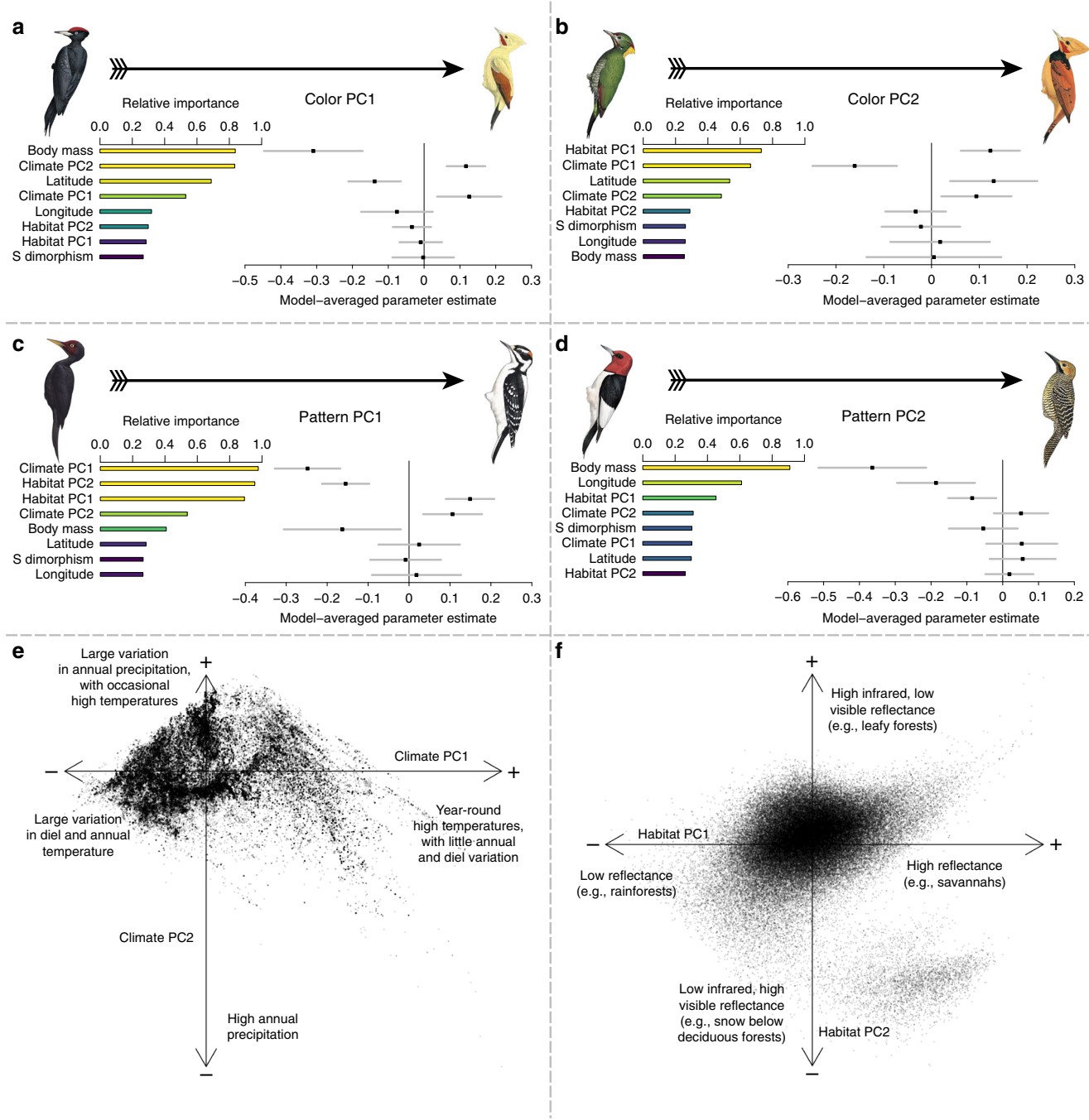

**Fig. 5** Variable importance scores and model-averaged parameter estimates from phylogenetic generalized least squares regressions. These quantify how colour and pattern vary as a function of climate, habitat, body mass, sexual size dimorphism, latitude and longitude, with summaries of the climate and habitat principal component analyses (PCA). Model-averaged $p$-values of explanatory factors are colour-coded from yellow to blue; only factors with $p$-values < 0.05 are coloured yellow and discussed here. **a** Dark birds are heavier and occur in wetter climates. **b** Greenish (as opposed to reddish) birds are found in more open habitats. **c** Less-patterned birds are found in aseasonal climates, open habitats, and temperate forests. **d** Birds patterned in large plumage elements, such as large colour patches, tend to be larger in body size. **e** Climate PCA results, illustrating the distribution of woodpeckers in climate space, with qualitative descriptions of the first two PC axes. **f** Habitat PCA results, showing the distribution of woodpeckers across global habitats, with qualitative descriptions of the first two PC axes. Illustrations © HBW Alive/Lynx Edicions

below, most analyses were run across this set of complete credible trees. However, for other analyses, and particularly for visualization purposes, we also derived a maximum clade credibility tree from this set of complete trees[43]. Finally, for each taxon in the complete tree, we identified the illustration that best represented it in the Handbook of the Birds of the World Alive. When the latter recognized multiple subspecies for a given taxon from the final tree, we used the nominate subspecies as our unit of analysis for colour and pattern (see below).

**Quantifying plumage colour and pattern from illustrations**. We calculated plumage colour and pattern scores for males of 230 species of woodpeckers using digital images of colour plates obtained from *The Handbook of the Birds of the World Alive*[17]. Each image was imported to Adobe Photoshop (Adobe Inc. San Jose, CA) at 300 dots per inch, scaled to a uniform size, and saved as a Tagged Image File (.TIF). Following creation of .TIF files, we ran a custom macro in ImageJ[44] to sample the red (R), green (G), and blue (B) pixel values for each of

1000 random, 9-pixel-diameter circles from each woodpecker image. RGB values were transformed to CIELAB coordinates, which is an approximately perceptually-uniform colour space (distance between points is perceptually equivalent in all directions)[45,46]. To calculate pairwise colour dissimilarity scores, we plotted the 1000 colour measurements from the first species (e.g., species A) in three-dimensional CIELAB space, as well as the 1000 measurements for the second species (e.g., species B) in the dyadic comparison. We then calculated the average Mahalanobis distance[47] between the colours representing species A and the colours representing species B. We repeated this process for every possible combination (26,335 unique dyadic combinations) to generate an overall colour dissimilarity matrix. Additionally, to facilitate a more in-depth investigation of the underlying variation in colour among species, as well as how such variation is related to environmental, genetic, and geographic influences we conducted principal components analysis (PCA) on all 230,000 colour measurements. Following PCA, we averaged principal component (PC) scores for each species to create mean PC scores describing the average colour values for each species (Fig. 2). PC1 describes a dark-to-bright continuum, as well as a blue-to-yellow continuum (high loadings for L* and b*; Supplementary Table 1; Fig. 2), while PC2 primarily describes a red-to-green continuum (high loadings for a*; Supplementary Table 1; Fig. 2).

We conducted pattern analyses on the same, scaled .TIF files for each species in ImageJ[44]. First, we split each image into R, G, B slices and then used the G layer for pattern analysis because this channel corresponds most closely to known avian luminance channels[48,49], which is thought to be primarily responsible for processing of pattern information[19,20]. We then used the Image Calibration and Analysis Toolbox[50] in ImageJ to conduct granularity-based pattern analysis. In this process, widely used to study animal patterning[51–54], images are Fast Fourier band pass filtered into a number of granularity bands that correspond to different spatial frequencies. For each filtered image, the "energy" at that scale is quantified as the standard deviation of filtered pixel values and corresponds to the contribution to overall appearance from pattern elements of that size. Pattern energy spectra were calculated for each species in a comparison across 17 bandwidths (from 2 pixels to 512 pixels, by multiples of √2), which we used for both pairwise pattern comparisons (pattern maps can be created to visualize differences; Fig. 3a), and to categorize overall plumage pattern with PCA (Fig. 3c, d). Pattern difference values were calculated by summing absolute differences between energy spectra at each bandwidth[50]; after principal components analyses, the first three PCs explained ~93% of the variance in overall pattern energy (Fig. 3c; Supplementary Table 5) across species (Fig. 3d). Pattern PC1 has large positive loadings for most element sizes/granularities, indicating that species with high PC1 scores have numerous pattern elements of different sizes (whereas species with low PC1 are relatively homogenous and with little overall patterning; Fig. 3d). Pattern PC2 has large positive loadings for small pattern elements, and negative loadings for intermediate and large pattern sizes so that species with high PC2 scores have lots of small pattern components, and species with low PC2 scores have more intermediate and large pattern size contributions (Fig. 3d).

As a check on our overall results, and to provide added insights into the factors driving plumage evolution in different regions of the body, we manually traced three regions of the body in the HBW illustrations and separated these into sets of images corresponding to: the back, including the wing and tail; the head, including the neck; and the breast and belly. We sent these separated illustrations through the same colour and pattern analytical pipeline described above. The ensuing colour and pattern spaces, and their associated loadings, are presented in Supplementary Figs. 10–18. The extent to which these different regions of the body function as independent plumage modules is questionable—plumage evolution in one region of the body is likely correlated with that in others. Hence, while we consider these analyses to offer some scientific insight into plumage evolution, we emphasize that the whole-body plumage analyses represent our preferred set of results. Future work would do well to study correlated plumage evolution across different regions of the body[55], and some work is now being undertaken in that research area[56].

**Photographic quantification of plumage colour and pattern**. To validate the use of colour plates for quantifying meaningful interspecific variation in plumage colour and pattern among woodpeckers, we employed digital photographic and visual ecology methods to quantify the appearance of museum specimens and compared these results to those obtained using the whole-body colour plates. Specifically, we used ultraviolet and visible spectrum images to create standardized multispectral image stacks and then converted these multispectral image stacks into woodpecker visual space. Photos were taken with a Canon 7D camera with full-spectrum quartz conversion fitted with a Novoflex Noflexar 35 mm lens, and two Baader (Mammendorf, Germany) lens filters (one transmitting only UV light, one transmitting only visible light). We took profile-view photograph pairs (one visible, one UV) under full-spectrum light (eyeColor arc lamps, Iwasaki: Tokyo, Japan, with UV-coating removed), then converted these image stacks into woodpecker visual space using data from *Dendrocopos major*[57] and average visual sensitivities for other violet-sensitive bird species[58]. The inferred peak-sensitivity (λmax) for the short-wavelength sensitive 1 (SWS1) cone of Great Spotted Woodpeckers, based on opsin sequence, is 405 nm[57]. After generating images corresponding to the quantum catch values (i.e., stimulation of the different photoreceptor types), we performed granularity-based pattern analyses with the Image Calibration and Analysis Toolbox[50] in ImageJ[44] using the image corresponding to the stimulation

of the avian double-cone, responsible for luminance detection[48,49], because this photoreceptor type is assumed to be involved in processing pattern information from visual scenes[19,20]. Additionally, because relative stimulation values do not generate perceptually-uniform colour spaces[59,60], we implemented visual models[61] to generate Cartesian coordinates for the colour values from each of 1000 randomly selected, 9-pixel diameter circles for each specimen and viewpoint (as we did with colour plates). Cartesian coordinates in this perceptually-uniform woodpecker colour space were then used to calculate pair-wise Mahalanobis distances[47] for each dyadic combination of measured specimens.

As with our colour plate-based analysis, we Z-score transformed colour and pattern distances (mean = 0, SD = 1), then combined these distances to create a composite plumage dissimilarity matrix incorporating overall plumage colour and pattern. Based on specimens available at the Cornell University Museum of Vertebrates, we endeavoured to measure up to three male specimens from at least one species of every woodpecker genus. We were able to measure 56 individuals from 23 woodpecker species (Supplementary Table 9). To compare the museum-based results to those from the colour plates, we derived species-level pairwise distances. We did so by finding the mean plumage distance between all specimens of one species and all of those of another, and repeating for all possible species pair comparisons. We repeated this process for both the colour only dissimilarity matrix, and the combined colour and pattern matrix. We subset the larger, plate-based colour-only and colour-plus-pattern matrices to the corresponding species, and compared the relevant matrices with Mantel tests. Our results from the museum specimens substantiated those from the illustrations—we found close correlations between colour dissimilarity (measured from specimens vs. measured from illustrations; Mantel test, $r = 0.74$, $p < 0.001$) and overall plumage dissimilarity (measured from specimens vs. measured from illustrations, Mantel test, $r = 0.72$, $p < 0.001$).

**eBird data management, curation, and analysis**. On 24 November 2017 we queried the eBird database for all records of each of the 230 species in our final woodpecker phylogeny. We excluded records for which we had low confidence in the associated locality information. Specifically, we excluded: (1) historical records, which are prone to imprecise locality information and are not associated with effort information, (2) records from (0°, 0°), (3) records that were considered invalid after review by a human (thus, flagged but unreviewed records were included), and (4) records that came from transects of longer than 5 km in length. Because eBird has grown exponentially in recent years, we connected directly to the database to ensure maximal data coverage for infrequently reported species. We made this analytical decision because the automatic filters that flag unusual observations can be imprecise in regions of the globe infrequently visited by eBirders; flagged observations remain unconfirmed (and not included in products such as the eBird basic dataset) until they are reviewed, and backlogs of unreviewed observations exist in some infrequently birded regions. This approach allowed us to increase our sample size for infrequently observed species. In contrast, other species are very well represented in the database. To reduce downstream computational loads, we used the R package *ebirdr* (https://github.com/eliotmiller/ebirdr) to downsample overrepresented species in a spatially stratified manner. Specifically, for each of the 230 species, we laid a grid of 100 × 100 cells over the species' extent, and randomly sampled and retained 60 points per cell. For most species, this had little to no effect, and fewer than 10% of points were thinned and removed from analysis; for a small number of well-sampled North American species, this excluded over 90% of points from analysis (Supplementary Data 1). In sum, this process reduced the original dataset from 13,513,441 to 1,037,628 records.

We used the R package *hypervolume*[62] to create pseudo-range maps around each species' point locations. Hypervolumes account for the density in the underlying points and can have holes in them, and are therefore much better suited to describing species' ranges than are, e.g., minimum convex polygons[62]. For every dyadic comparison (i.e., for every species pair comparison), we used *hypervolume* to calculate the Sørenson similarity index between the species' inferred geographic ranges. We summarized these similarities in a pairwise matrix, which we subsequently converted to a dissimilarity matrix such that a value of 1 represented complete allopatry (no overlap in geographic distributions), and a value of 0 represented perfect sympatry (complete overlap in geographic distributions).

We used the *raster* package to match each species' point locations to climatic values using WorldClim bioclimatic data[63]. These data describe the annual and seasonal climatic conditions around the globe. After querying species' climatic data, we bound the resulting files together and ran a single large correlation matrix PCA across all climate variables except bio7, which is simply the difference between bio5 and bio6. We retained species' scores along the retained PC axes and used scores along the first two PC axes to calculate species-level hypervolumes in climate space. These first two axes explained 85% of the variance in the climates occupied by woodpeckers. The first axis described a gradient from places that are generally warm throughout the year, to areas that show seasonal variation in temperature and large diurnal shifts in temperature. The second axis described a gradient from areas that receive precipitation in seasonal pulses, have some hot months and have large swings in temperature over the course of a day, to areas that always receive lots of rain. Again, for each dyadic comparison, we calculated a Sørenson similarity index, and then converted the resulting values to a dissimilarity matrix.

**Querying habitat data**. We used *ebirdr*, which harnesses GDAL (http://www.gdal.org/), to bind species' point locations into ~50 MB-sized tables, then converted the resulting tables into KML (Keyhole Markup Language) files, which we uploaded and converted into Google Fusion Tables (https://fusiontables.google.com). This particular file size was chosen after we employed a trial-and-error process to determine the most efficient query size for Google Earth Engine (see below). Once accessible as a Fusion Table, we fed the tables into custom Google Earth Engine scripts. For every eBird observation, these scripts identified the MODIS satellite reflectance values[64] from the observation location within a 16-day window of the observation. We queried data specifically from the MODIS MCD43A4 Version 6 Nadir Bidirectional reflectance distribution function Adjusted Reflectance (NBAR) data set, a daily 16-day product which "provides the 500 m reflectance data of the MODIS 'land' bands 1–7 adjusted using the bidirectional reflectance distribution function to model the values as if they were collected from a nadir view" (https://lpdaac.usgs.gov/node/891). At the time of query, this dataset was available for the time period 18 February 2000 to 14 March 2017, which corresponded to the time period in which most of our eBird records were recorded. The year of all other records was adjusted up or down to fall within the available satellite data, e.g., observations from 10 November 2017 became 10 November 2016. This method is appealing in that it incorporates species' spatiotemporal variation in habitat availability and use, although for most woodpecker species such variation is minimal.

After querying species' habitat data, we downloaded and combined the resulting files from Google Earth Engine, dropping any records that were matched to incomplete MODIS data. *ebirdr* contains functions to automatically combine and process these files from Google Earth Engine (although the functions currently employ Google Fusion Tables, which will be discontinued in December 2019). We then ran a single large correlation matrix PCA across all 7 MODIS bands. Before doing so, we natural log-transformed bands 1, 3, and 4, as a few extreme values along these bands hampered our initial efforts to ordinate this dataset. We retained the first two PC axes, which explained 81% of the variance in the habitats occupied by woodpeckers. The first described a gradient from closed forests to open, reflective habitats. The second described a gradient from regions with high visible and low infrared reflectance to those with low visible and high infrared reflectance. This dichotomy is used to identify snow in MODIS snow products (https://modis.gsfc.nasa.gov/data/atbd/atbd_mod10.pdf). Thus, at the species-average level, the second habitat PC axis functionally described a gradient between seasonally snow-covered (temperate) forests and tropical woodland. Again, for each dyadic comparison, we calculated a Sørenson similarity index, and then converted the resulting matrix to a dissimilarity matrix.

**Multiple distance matrix regression**. After the steps described above, we had data from four variables hypothesized to explain plumage variation across woodpeckers in the form of four pairwise distance matrices: genetic distances, climate dissimilarity, habitat dissimilarity, and geographic range dissimilarity. We combined the plumage colour and the plumage pattern dissimilarity matrices into a single matrix by independently standardizing each using *z*-scores, then calculating the element-wise sum of each dyadic comparison. We then related the four explanatory matrices to the single dependent plumage dissimilarity matrix using multiple distance matrix regression, with 999 permutations[65]. To account for phylogenetic uncertainty, we iterated this process over each of the 1000 complete phylogenies. The resulting model was highly significant (multiple distance matrix regression, median *p* across all complete phylogenies = 0.030), but fairly low in explanatory power (median *r* = 0.170), reflecting the massive variation incorporated into these five 230 × 230 matrices. It bears emphasizing that this correlation coefficient represents not, e.g., the degree to which two variables are correlated, but rather the degree to which the dissimilarity between various clouds of points can explain the dissimilarity in other clouds of points; low explanatory power is to be expected. Three of the four dissimilarity matrices were significantly and positively associated with plumage dissimilarity: increasing genetic distance (multiple distance matrix regression, median *p* = 0.020), climate dissimilarity (median *p* = 0.007), and habitat dissimilarity (median *p* = 0.006) all lead to increasing plumage dissimilarity. The distributions of correlation coefficients across the cloud of credible trees for these explanatory variables are shown in Supplementary Fig. 1. In this analysis, geographic range dissimilarity was not significantly associated with plumage dissimilarity.

**Modified Mantel correlogram**. The likelihood that sympatric species evolve plumage mimicry is not thought to be monotonically related to range overlap. Instead, hypotheses to explain plumage mimicry propose that only certain species pairs that are both closely sympatric and ecologically similar will converge dramatically in appearance[14]. Thus, we did not expect a continuous relationship between geographic range dissimilarity and plumage dissimilarity; rather, we expect a threshold relationship wherein plumage convergence occurs in dyads with high geographic overlap. We therefore implemented a modified Mantel correlogram approach to test whether such a threshold existed[66]. For this, we manually created a series of matrices where we converted all elements in the plumage dissimilarity matrix to values of 1, except for dyads with plumage dissimilarity scores in a certain range. Specifically, the dissimilarity scores within a given range for a given analysis (ranges tested: 0–0.1, 0.1–0.2, 0.2–0.3, 0.3–0.4, 0.4–0.5, 0.5–0.6,

0.6–0.7, 0.7–0.8, 0.8–0.9, and 0.9–1) were set to a value of 0, and all other dissimilarity scores were set to a value of 1. We then sequentially input these matrices as the dependent variable into the same multiple distance matrix regression described above, repeatedly calculating the significance and partial correlation coefficient of the geographic range dissimilarity matrix with that of plumage dissimilarity. This approach allowed us to examine how the correlation between plumage and range dissimilarities varied across a range of plumage dissimilarities, while simultaneously incorporating the influences of evolutionary relationships, and climate and habitat dissimilarities.

We found that geographic range dissimilarity was significantly associated with plumage dissimilarity for the most similar looking species pairs (Fig. 4). Dyadic comparisons with plumage dissimilarities of 0–0.2 include such pairs as *Dryobates pubescens* and *Dryobates villosus* (purported plumage mimics), *Gecinulus grantia* and *Blythipicus pyrrhotis* (which are quite similar looking), and *Picus awokera* and *Melanerpes striatus* (not closely similar, but do share colour and pattern elements). Put differently, geographic range overlap per se is statistically significantly associated with increasing plumage phenotype matching between already similar looking species pairs. Notably, the relationship was reversed at intermediate levels of plumage dissimilarity; geographic range overlap is statistically significantly associated with decreasing plumage phenotype matching between somewhat similar looking species pairs (plumage dissimilarities of 20–30%). Dyadic comparisons with dissimilarities in this range include *Campethera abingoni* and *C. maculosa* and *Veniliornis spilogaster* and *Celeus obrieni*. Although it is true that this signal could be interpreted as evidence that allopatry in and of itself drives plumage divergence between somewhat similar looking species pairs, this seems biologically implausible. A more likely reason for this signal is substantial plumage differentiation between pairs of birds at intermediate levels of sympatry[67]. The fact that some degree of sympatry is associated with rapid plumage divergence is expected by theory[68], and is likely due to strong selection to avoid unsuccessful hybridization (i.e., reinforcement), or to avoid accidentally targeting heterospecifics for aggression[69]. Whether the relaxation of plumage divergence in closer sympatry could be attributed to shared habitats or climates, or to some other selective pressure, was heretofore unknown[67]. We show that in woodpeckers, after accounting for other likely selective forces, either one or both of the species in pairs that have attained close sympatry may evolve towards the phenotype of the species with which they co-occur.

To further assess the strength of this striking result, we devised a simulation to determine whether such a pattern might result from chance alone. To do so, we repeatedly derived five matrices with the same intercorrelations among them as our observed dependent (plumage dissimilarity) and four independent matrices (genetic, habitat, climate, and range dissimilarity). When input into the multiple distance matrix regression described above, the resulting matrix-specific coefficients and overall power of the simulated independent matrices to explain variation in the simulated woodpecker plumage dissimilarity matrix was identical to that in the observed matrices. By using these same matrices in the modified Mantel correlogram approach described above, we were able to test whether the pattern observed in Fig. 4 (red line) could result by chance alone. After 200 iterations of the simulation, we calculated the standardized effect size of the correlation coefficient of each thresholded plumage dissimilarity matrix with range dissimilarity as the difference between the observed value and the mean of the simulations, divided by the standard deviation of the simulated correlation coefficients. Standardized effect sizes greater than $+/-1.96$ reflect observed correlation coefficients that deviated beyond 95% of simulated values. These simulations strongly support our finding that close sympatry—above and beyond evolutionary relatedness, shared climate, and shared habitat preferences—drives otherwise unexpectedly high levels of plumage convergence in woodpeckers. In short, close sympatry appears to be associated with occasional plumage mimicry in woodpeckers. We recognize that Mantel tests, and presumably by extension variations such as that described here, can suffer from inflated type I error rates[70,71]. Future work should seek to further establish the relevance of sympatry to driving plumage mimicry in birds with alternative approaches.

**Identification of putative plumage mimics**. We developed a method to identify high-leverage dyadic comparisons in Mantel tests and multiple distance matrix regressions. We used this to identify species pairs that have converged above and beyond that expected by shared climates and habitats. The process works as follows. In the first step, the observed correlation statistic is calculated. In our case, that was the correlation coefficient of a thresholded plumage dissimilarity matrix (values from 0–0.2 set to 0, all others set to 1) with the geographic range dissimilarity matrix. The statistic can also be the correlation coefficient from a regular or partial Mantel test; we confirmed that the method yielded similar results when we employed it with a partial Mantel test between the continuous plumage dissimilarity matrix, geographic range dissimilarity, and genetic distance. In the second step, each element (dyad) in the relevant matrix is modified in turn, and the relevant correlation statistic calculated and retained after each modification. We tested three methods of modifying dyads, i.e., three different approaches to this second step. All yielded similar results. (A) The value can be randomly sampled from the off-diagonal elements in the matrix. (B) The value can be set to NA and the correlation statistic calculated using all complete observations. (C) For the thresholded matrix, the test element can be swapped for the other value; zeros

become ones, and ones become zeros. In the third step, only necessary for approach A, the process is iterated multiple times, and the modified correlation coefficient for every element, at each iteration, is stored as a list of matrices. In the fourth step, again only necessary for approach A, these matrices are summarized by taking the element-wise average. In the fifth step, the leverage of each dyad is calculated by subtracting the observed correlation statistic from each element in the averaged matrix. Finally, the matrix can be decomposed into a pairwise table and sorted by the leverage of each dyad. In our case, dyads that have high leverage, and are large contributors to the positive correlation between close plumage similarity and geographic range overlap, have the largest negative values (i.e., modifying their observed plumage dissimilarity score most diminished the observed positive correlation between range and plumage).

We used this method to identify the most notable plumage mimics across woodpeckers, after accounting for shared evolutionary history, climate, and habitat use. Many purported mimicry complexes were responsible, including the Downy-Hairy system[22], and repeated convergences between members of *Veniliornis* and *Piculus*[23], *Dinopium* and *Chrysocolaptes*, and *Dryocopus* and *Campephilus*[24]. Convergence between the Helmeted Woodpecker (*Dryocopus = Celeus galeatus*) and *Campephilus robustus* was also detected[15], as was convergence between members of *Thripias* and *Campethera*, *Meiglyptes* and *Blythipicus*, and *Hemicircus* and *Meiglyptes*.

**Phylogenetic least squares regression**. We derived species' average scores along the first two axes of a plumage colourPCA (Fig. 2), a plumage pattern PCA (Fig. 3), the climate PCA described above, the habitat PCA described above, and species' average latitude (absolute value) and longitude of distribution. Additionally, we mined body mass data from Dunning[72]. For those species for which mass was listed separately for males and females, we calculated sexual size dimorphism *sensu* Miles et al.[73]. These authors additionally reported dimorphism measures from a number of species not available in Dunning[72]. We then combined these datasets, resulting in sexual size dimorphism measures for 94 of 230 species. Sexual size dimorphism in woodpeckers is generally small compared to other avian groups such as the Icteridae, and they have not traditionally been considered a clade characterized by strong sexual selection pressures. During the process of combining datasets, we noticed that one of the most well-known of sexually size-dimorphic species, *Melanerpes striatus*, was characterized in both databases as having larger females than males. This is incorrect—males are notably larger than females—and we replaced the values with the midpoint of ranges given in ref. [17]. We used *Rphylopars*[74] to impute missing body mass and size dimorphism data, which we did using a Brownian motion model and the observed variance-covariance matrix between all traits except for plumage colour and pattern.

Treating climate, habitat, latitude, longitude, natural log body mass, and sexual size dimorphism as explanatory variables, we used multi-model inference to identify PGLS regression models that explained each of the four PCA plumage axes of interest. We also visualized pairwise correlations and distributions of these traits using *corrplotter*[75] (Supplementary Fig. 2). We used a model averaging approach to determine which explanatory variables strongly influenced plumage (Fig. 5). To test the robustness of our conclusions to phylogenetic uncertainty, for each dependent variable (colorPC1, colorPC2, patternPC1, and patternPC2), we identified all explanatory variables with model-averaged coefficients that did not overlap zero. We then fit a series of 1000 PGLS regressions per dependent variable to the identified variables where, for each regression, we used a different one of the complete phylogenies. Variation in the coefficient estimations was small, as shown in Supplementary Figs. 3–6. In the main text, when reporting pseudo-$r^2$ and values for the PGLS regressions, we report the median values from these 1000 models.

**Reporting summary**. Further information on experimental design is available in the Nature Research Reporting Summary linked to this article.

## Data availability
All data supporting the findings of this study are available within the paper and its supplementary information files.

## Code availability
All computer code necessary to run these analyses is available in the purpose-built R package *ebirdr*, available at https://github.com/eliotmiller/ebirdr.

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

## Acknowledgements

We thank T. Auer, D. Caetano, D. Fink, R. Grant, L. Harmon, W. Hochachka, K. Horton, S. Kelling, R. Maia, A. Patton, M. Pennell, M. Strimas-Mackey, D. Toews, J. Uyeda, C. Wood and R. Zenil-Ferguson for statistical, graphical, and intellectual input. E.T.M., G.M.L. and R.A.L. were supported by NSF grants 1402506, 1523748, and 1523895, respectively. E.T.M., G.M.L., R.A.L. and A.C.L. were supported by the Cornell Lab of Ornithology. B.G.F. was supported by Banting Canada (379958) and the Biodiversity Research Centre at the University of British Columbia.

## Author contributions

E.T.M., G.M.L., B.G.F., A.C.L. and R.A.L. were responsible for conceptualization and writing of the manuscript. E.T.M. and R.A.L. were responsible for analysis and visualization.

## Additional information

**Competing interests:** The authors declare no competing interests.

