## [Peer Review File · Nature Communications]

Reviewers' Comments:

Reviewer #1:

Remarks to the Author:

This study represents a very great deal of data manipulation on complex colour and pattern, and geographic range and climate data. On superficial reading, this is a technical tour de force and I believe this is why it was sent to a general journal such as Nature Communications. The authors find significant (if very weak) signals of woodpecker plumage variation (both colour and pattern) across species each being independently correlated with climate and habitat. Convergence in plumage based on broad habitat and latitude seems to have been expected based on previous, less quantitative work, and so it is gratifying to see it confirmed with these large citizen science and digitized plumage colouration datasets.

However, I believe the main new claim is that, after controlling for climate and habitat, pairs of species that are very similar in plumage have ranges that overlap, on average, more than pairs of species that are less similar or even dissimilar in plumage, implying selection for plumage mimicry in sympatric woodpeckers. If this were true, it would be a result with broad general natural history interest. This result is based on a positive partial distance correlation between plumage dissimilarity (with pairs scored as not dissimilar vs. slightly or very dissimilar, a pseudo-binomial character) and range dissimilarity score (which is scored continuously). They further claim that pairs of species with intermediate plumage similarity have "partial but incomplete range overlap." This latter claim gave me pause, and I explain what I think they did here. It is very possible I've got this wrong:

The authors scored pairs with 0.2-0.3 plumage dissimilarity scores as minimally dissimilar (dissimilarity set to 0), and all other dissimilarity pair-values (so, both 0-0.2 and greater than 0.3) as maximally dissimilar (dissimilarity set to 1), and found a negative partial correlation with range dissimilarity, ie species pairs in the 0.2-0.3 dissimilarity range had less range overlap than pairs in the 0-0.2 and >0.3 plumage dissimilarity range. But these 0.2-0.3 plumage dissimilarity pairs, scored as having dissimilarity = 0 could generally be allopatric (range dissimilarity = 1) with a mix of overlaps for the other two groups and one would get a partial negative correlation, no? Indeed, we already 'know' that the 0-0.2 plumage dissimilarity pairs are tend to have high range overlap.

I apologize if I have misinterpreted this all-important test, but perhaps if I cannot puzzle it through, others might not either. Perhaps all that is needed is a bit more explanation.

The more general concern with this study is that the R^2 values are never more than 1% of explained variation for the proposed mimicry patterns. These distance analyses include phylogenetic correction as well as habitat and climate correlations (which are likely to covary extremely tightly with sympatry), and so I am willing to believe (I have little intuition for partial Mantel tests) that the patterns are not clearly biased by a few well-known species-pairs, However, it is hard to know the extent to which generalities can be made with such small effect sizes. The general pgl models looking at the PCAs of plumage and climate and habitat have more reasonable r-squares, but if the paper's potential to capture folks' attention is indeed the results depicted in Extended Data Figure 1, I am not sure how the authors should proceed here.

I have made a series of suggestions and asked questions directly on the manuscript using "track changes" in Word.

Reviewer #2:

Remarks to the Author:

This paper provides a well-written and highly interesting account of what in my view amounts to an extraordinary study. The authors have integrated a number of state of the art methods to resolve some of the key drivers of plumage evolution in the woodpeckers. My suggestions below are relatively minor and are provided in the hope they may help the authors make some improvements to the paper.

One of the most interesting biological findings of the paper is the identification of selection promoting mimicry at a broad taxonomic scale. Did you try and get the word mimicry in the title? My view is that this key aspect of the study gets lost with the term "geographic range overlap".

I think that at times you overreach with your conclusions, by overgeneralizing them. It is important to keep in mind that this work describes patterns in a single clade (Google is telling me that woodpeckers are a subfamily) – but woodpeckers are not even mentioned in the title or in the abstract. Your title suggests (subtly) that you have id'd the drivers of plumage evolution in general but it is important to recognize that other drivers, like sexual selection, will likely be more important in other groups, so I think it is worth being more clear about that. Your study sets up the opportunity to do similar research in other groups and it will be fascinating to contrast similarities and differences.

On that note, one reason you did not find much of a signal for sexual selection could be that variance in sex-biased direction sexual selection might simply be low in the woodpeckers as a whole. Moreover sexual size dimorphism as your single index of sexual selection is crude at best. Is there even that much variation in SSD in the group as a whole? What about mating system? Are there any non-monogamous woodpeckers? Do woodpeckers have much extra-pair paternity? It might be worth touching on these points in your paper.

The figures are excellent.

Reviewer #3:

Remarks to the Author:

"Climate, habitat, and geographic range overlap drive plumage evolution" reports on a interesting set of results and an impressive amount of work on woodpecker data. Overall, the manuscript provides interesting insights, although some claims made by the authors seem self-aggrandizing and unnecessary. For example, the title and the language throughout the paper imply that the results are generally applicable but the data are clearly constrained to a single narrow clade of birds, which, for various reasons may not reflect general Class-wide patterns, let alone, patterns in other taxa. In my opinion, being specific about the fact that these findings apply only to woodpeckers (e.g., by adding "in woodpeckers" to the title) does not diminish the validity of this contribution, especially because the clade in question is both interesting in itself and globally distributed. Other unnecessary claims include that "to date, no study has taken an integrative approach to understand how these factors determine the evolution of colour and pattern across a large clade of organisms" (I can think of several recent studies on sexual selection that have attempted to do just that with MUCH larger samples of species than the 230 spp in this study), or that this is the "first demonstration of selection promoting the evolution of mimicry at a broad phylogenetic scale" (what about the impressive body of work on butterflies? What about phylogenetic comparative studies that have analyzed acoustic mimicry in birds?), or describing a correlation as "tight" when r is only 0.74. In any case, distractions aside, I see value in this contribution and I like some of the methodological approaches used (BTW: there again I find the authors take it a bit too far in claiming novelty of their methods given that partial Mantel tests are commonplace in the phylogenetic comparative literature). Nevertheless, I also worry about other

methodological choices made by the authors and their potential effects on the reported results. Specifically...

- Colour data: Bird guides often depict birds in positions that emphasize their defining characteristics. Thus, choosing a random set of 1000 points from drawings that do not cover the totality of a specimen could generate significant biases. In my opinion, it would be MUCH more informative to use instead landmarks like the ones used in geometric morphometrics when collecting colour data. By standardizing the positions where you sample colour you may be better able to assess similarities and differences among species and may be less prone to the inherent biases of sampling from artists' depictions of the spp in the clade of interest. I am honestly not quite convinced by the information provided that the approach taken accurately measures what the authors are after (even though I can see that it is consistent with metrics obtained from photographs of specimens... basically because photographs suffer from similar issues of perspective and coverage).

- Spatial data: while eBird is amazing for many reasons, it does not seem like the most natural data source for species distribution ranges given that, as the authors acknowledge, eBird coverage varies regionally and there are issues with quality control in regions of the world that are poorly visited by eBirders. The authors therefore are forced to go through a frankly quite complicated set of procedures to collate and distill eBird data into convex hull polygons depicting each species' range, sometimes having to sacrifice quality (e.g., allowing flagged but not reviewed sightings) to increase sample size. Given that what the authors are after are basic descriptions of where a species is likely to be found, it is entirely unclear to me why they do not use instead the free range maps provided by BirdLife international, which, while admittedly not perfect, are the current standard for distribution data in this group. It would be at least nice to know that the assumptions and procedures involved in generating species distributions from spotty eBird data do not result in any meaningful differences with respect to what one would find when using birdLife maps instead.

- Phylogenetic uncertainty: It is clear that while a recent phylogeny for woodpeckers is available, there are still some issues with phylogenetic uncertainty. Instead of embracing such uncertainty, the authors appear to go through great lengths to reduce it and end up apparently working with a single phylogenetic hypothesis that is derived after applying multiple assumptions and approximations. Here again, it would be nice to see that this is not unduly influencing what we see. At the very least, it would be nice to apply all procedures used in this study to a large sample of trees from the posterior distribution of trees from the Shakya et al. study and report the level of congruence seen across them. The birdtree.org posterior may also be of use...

- Intraspecific and individual variation: The approach taken here also completely ignores within-species variation by sampling only one "specimen" (i.e., one artist's depiction) per species and explicitly ignoring all but the nominate race, when more than one is available. Nominate subspecies are not always the most widely distributed and the possibility of within-species geographic variation seems important to consider when analyzing the effects of range overlap. I find it hard to believe that if this study was on any other morphological trait, we would be content with having a single measurement per species! I can't remember if the authors consider variation in coloration among the sexes either...

- Sexual selection. Many studies have pointed out to a tradeoff between size dimorphism and sexual dichromatism in birds. Furthermore... many of the most sexually selected birds we know are dichromatic but not sexually dimorphic. This manuscript uses only size dimorphism as a metric sexual selection and therefore cannot really make any general statements about the role or lack thereof sexual selection on the processes being studied. At the very least some discussion on the topic would be warranted.

Minor comments:

- The paragraph starting in ln 65 talks about effects of total precipitation on how dark species are and how they are patterned. However, the paragraph starting in line 85 talks again about darkness (colorPC1) but mentions that the best model for it included only body mass, latitude, and seasonality in precipitation... what am I missing? Was total precipitation not part of this best model? Why then were two different models used to analyze this variable. Apologies if I am missing something obvious but the way this is currently written makes it very difficult to understand which results come from different analyses. A similar issue is observed for pattern variables.

We offer our heartfelt thanks to the editors and the reviewers for their numerous insights and time spent reading and handling our manuscript. After conceiving of, implementing, and running additional analyses, we have further corroborated the strength of our results, and we thank the reviewers for the impetus to do so. Accordingly, we have now revised the manuscript to reflect these checks on our work, and we have revised to reflect reviewer comments on specific aspects of the manuscript. We respond directly to these comments below. Our responses are in blue text immediately below the original reviewer comment. Thank you again for your comments, which greatly improved the manuscript.

Reviewer 1

However, I believe the main new claim is that, after controlling for climate and habitat, pairs of species that are very similar in plumage have ranges that overlap, on average, more than pairs of species that are less similar or even dissimilar in plumage, implying selection for plumage mimicry in sympatric woodpeckers. If this were true, it would be a result with broad general natural history interest. This result is based on a positive partial distance correlation between plumage dissimilarity (with pairs scored as not dissimilar vs. slightly or very dissimilar, a pseudo-binomial character) and range dissimilarity score (which is scored continuously). They further claim that pairs of species with intermediate plumage similarity have “partial but incomplete range overlap.” This latter claim gave me pause, and I explain what I think they did here. It is very possible I've got this wrong:

The authors scored pairs with 0.2-0.3 plumage dissimilarity scores as minimally dissimilar (dissimilarity set to 0), and all other dissimilarity pair-values (so, both 0-0.2 and greater than 0.3) as maximally dissimilar (dissimilarity set to 1), and found a negative partial correlation with range dissimilarity, ie species pairs in the 0.2-0.3 dissimilarity range had less range overlap than pairs in the 0-0.2 and >0.3 plumage dissimilarity range. But these 0.2-0.3 plumage dissimilarity pairs, scored as having dissimilarity = 0 could generally be allopatric (range dissimilarity = 1) with a mix of overlaps for the other two groups and one would get a partial negative correlation, no? Indeed, we already ‘know’ that the 0-0.2 plumage dissimilarity pairs are tend to have high range overlap.

I apologize if I have misinterpreted this all-important test, but perhaps if I cannot puzzle it through, others might not either. Perhaps all that is needed is a bit more explanation.

>>We thank the reviewer for their careful consideration of our partial Mantel correlogram approach. The reviewer was correct in that the dyads driving the negative correlation at intermediate levels of plumage similarity could have simply been allopatric, and in general the analysis is somewhat unintuitive. To help readers to better understand this critical result, we have carefully elaborated on the results throughout this section (lines 445-467) of the manuscript, including detailing the important allopatry point raised by the reviewer. These additions provide valuable clarification of the results, and we again thank the reviewer for their insights here.

The more general concern with this study is that the R^2 values are never more than 1% of explained variation for the proposed mimicry patterns. These distance analyses include phylogenetic correction as well as habitat and climate correlations (which are likely to covary extremely tightly with sympatry), and so I am willing to believe (I have little

intuition for partial Mantel tests) that the patterns are not clearly biased by a few well-known species-pairs, However, it is hard to know the extent to which generalities can be made with such small effect sizes. The general pgl models looking at the PCAs of plumage and climate and habitat have more reasonable r-squares, but if the paper's potential to capture folks' attention is indeed the results depicted in Extended Data Figure 1, I am not sure how the authors should proceed here.

>>The reviewer is correct that the reported r^2 from the multiple distance matrix regression is small. We initially shared these concerns. However, an unintuitive point is that this r^2 value is not analogous to that from, for example, a standard OLS regression. Instead, this measures the degree to which the overlaps between clouds of points in multivariate space can explain the overlaps between another set of points in multivariate space. There is a massive amount of variation encompassed in these clouds of points, and finding a statistically significant, biologically plausible signal is, despite the low r^2 values, a strong point of the paper. We have added language to address our explanatory power more explicitly (lines 413-419).

I have made a series of suggestions and asked questions directly on the manuscript using "track changes" in Word.

>>We thank the reviewer for their excellent suggestions throughout the attached document, and we have revised our writing to incorporate all of these edits (e.g., lines 185, 435, and 441).

Reviewer 2

One of the most interesting biological findings of the paper is the identification of selection promoting mimicry at a broad taxonomic scale. Did you try and get the word mimicry in the title? My view is that this key aspect of the study gets lost with the term "geographic range overlap".

>>This is a good suggestion, and we have modified the title to do so.

I think that at times you overreach with your conclusions, by overgeneralizing them. It is important to keep in mind that this work describes patterns in a single clade (Google is telling me that woodpeckers are a subfamily) – but woodpeckers are not even mentioned in the title or in the abstract. Your title suggests (subtly) that you have id'd the drivers of plumage evolution in general but it is important to recognize that other drivers, like sexual selection, will likely be more important in other groups, so I think it is worth being more clear about that. Your study sets up the opportunity to do similar research in other groups and it will be fascinating to contrast similarities and differences.

>>We agree that some language in our first submission was too general, so we have taken care to specify throughout the manuscript that our results are, until proven otherwise, specific to woodpeckers (e.g., in the title, lines 15, 58, etc.). We also now point to the value of investigating these questions in other groups in the final paragraph of the conclusion.

On that note, one reason you did not find much of a signal for sexual selection could be that variance in sex-biased direction sexual selection might simply be low in the woodpeckers as a whole. Moreover sexual size dimorphism as your single index of sexual selection is crude at best. Is there even that much variation is SSD in the group as a whole? What about

mating system? Are there any non-monogamous woodpeckers? Do woodpeckers have much extra-pair paternity? It might be worth touching on these points in your paper. >>The reviewer raises an important point. The answer to both of these questions is “probably not”, and we have added language to caveat this aspect of our investigation (lines 535-537).

Reviewer 3

“Climate, habitat, and geographic range overlap drive plumage evolution” reports on a interesting set of results and an impressive amount of work on woodpecker data. Overall, the manuscript provides interesting insights, although some claims made by the authors seem self-aggrandizing and unnecessary. For example, the title and the language throughout the paper imply that the results are generally applicable but the data are clearly constrained to a single narrow clade of birds, which, for various reasons may not reflect general Class-wide patterns, let alone, patterns in other taxa. In my opinion, being specific about the fact that these findings apply only to woodpeckers (e.g., by adding “in woodpeckers” to the title) does not diminish the validity of this contribution, especially because the clade in question is both interesting in itself and globally distributed. Other unnecessary claims include that “to date, no study has taken an integrative approach to understand how these factors determine the evolution of colour and pattern across a large clade of organisms” (I can think of several recent studies on sexual selection that have attempted to do just that with MUCH larger samples of species than the 230 spp in this study), or that this is the “first demonstration of selection promoting the evolution of mimicry at a broad phylogenetic scale” (what about the impressive body of work on butterflies? What about phylogenetic comparative studies that have analyzed acoustic mimicry in birds?), or describing a correlation as “tight” when r is only 0.74. In any case, distractions aside, I see value in this contribution and I like some of the methodological approaches used (BTW: there again I find the authors take it a bit too far in claiming novelty of their methods given that partial Mantel tests are commonplace in the phylogenetic comparative literature).

>>We agree that at times the language in our first submission was too general, so we have taken care to specify throughout the manuscript that our results are specific to woodpeckers (e.g., in the title, lines 15, 58, etc.). We have removed most mentions of novelty, but we maintain that the methods and results represent important advances.

Colour data: Bird guides often depict birds in positions that emphasize their defining characteristics. Thus, choosing a random set of 1000 points from drawings that do not cover the totality of a specimen could generate significant biases. In my opinion, it would be MUCH more informative to use instead landmarks like the ones used in geometric morphometrics when collecting colour data. By standardizing the positions where you sample colour you may be better able to assess similarities and differences among species and may be less prone to the inherent biases of sampling from artists' depictions of the spp in the clade of interest. I am honestly not quite convinced by the information provided that the approach taken accurately measures what the authors are after (even though I can see that it is consistent with metrics obtained from photographs of specimens... basically because photographs suffer from similar issues of perspective and coverage).

>>We agree with the reviewer that using landmarks can be insightful and important, and this approach was the focus of our initial investigation into the group. However, what we found in practice was that landmarks left too much open to interpretation. Plumage patches do not necessarily map identically across species—does one add the landmark on the patch that the human identifies as important (e.g., the malar stripe), or does one always put it on the same place morphologically, even if it misses the patch the human identifies as important? Nevertheless, in this revision, we have now manually divided all 230 woodpecker illustrations into head, back, and belly portions of the birds, and analyzed each separately. Results are very similar to those from the whole-body analysis—which remains our preferred level of analysis—but they do offer some interesting insights into differing selective forces that may be operating on different parts of the body (e.g., lines 273-284). We thank the reviewer for encouraging us to revisit this approach.

Spatial data: while eBird is amazing for many reasons, it does not seem like the most natural data source for species distribution ranges given that, as the authors acknowledge, eBird coverage varies regionally and there are issues with quality control in regions of the world that are poorly visited by eBirders. The authors therefore are forced to go through a frankly quite complicated set of procedures to collate and distill eBird data into convex hull polygons depicting each species' range, sometimes having to sacrifice quality (e.g., allowing flagged but not reviewed sightings) to increase sample size. Given that what the authors are after are basic descriptions of where a species is likely to be found, it is entirely unclear to me why they do not use instead the free range maps provided by BirdLife international, which, while admittedly not perfect, are the current standard for distribution data in this group. It would be at least nice to know that the assumptions and procedures involved in generating species distributions from spotty eBird data do not result in any meaningful differences with respect to what one would find when using birdLife maps instead.

>>We contend that our original approach is superior for three reasons. First, the BirdLife maps do not match modern taxonomy. Using these would have forced us to follow one of three unfortunate analysis pipelines: dropping all species for which there was no BirdLife map, using the outdated birdtree.org phylogeny and forcing eBird taxa into these outdated taxonomic concepts, or manually splitting BirdLife maps based on our intuition. Second, the reviewer's statement that these "are the current standard for distribution data in this group" is somewhat subjective. In our experience, the BirdLife maps range from very good to significantly inaccurate. Also, because they are polygons, they inevitably overestimate species' geographic distributions (e.g., see this article covering a recent paper in *Biological Conservation* (<https://news.mongabay.com/2017/04/overestimated-range-maps-used-for-endemic-birds-in-indias-western-ghats-lead-to-underestimated-threats-study-finds/>)). Third, BirdLife maps are presence/absence only, whereas eBird and our hypervolume approach (not convex hull as stated by the reviewer) can account for density.

Phylogenetic uncertainty: It is clear that while a recent phylogeny for woodpeckers is available, there are still some issues with phylogenetic uncertainty. Instead of embracing such uncertainty, the authors appear to go through great lengths to reduce it and end up apparently working with a single phylogenetic hypothesis that is derived after applying multiple assumptions and approximations. Here again, it would be nice to see that this is not unduly influencing what we see. At the very least, it would be nice to apply all

procedures used in this study to a large sample of trees from the posterior distribution of trees from the Shakya et al. study and report the level of congruence seen across them. The birdtree.org posterior may also be of use...

>>We agree that, whenever possible, it is worthwhile to account for phylogenetic uncertainty in analyses. To that end, we contacted the lead author of the recent woodpecker phylogeny (Subir Shakya), who kindly provided us with a large set of credible trees. We used a randomly sampled subset of these trees (after dropping the burn-in period) to re-run our complete analyses 1,000 times across each of the complete phylogenies. The results are essentially identical to those from the original maximum clade credibility tree we used (Extended Data Figs. 1 and 3-6), but we are grateful that the reviewer suggested this, as it lends further support to the strength of our conclusions.

Intraspecific and individual variation: The approach taken here also completely ignores within-species variation by sampling only one “specimen” (i.e., one artist’s depiction) per species and explicitly ignoring all but the nominate race, when more than one is available. Nominate subspecies are not always the most widely distributed and the possibility of within-species geographic variation seems important to consider when analyzing the effects of range overlap. I find it hard to believe that if this study was on any other morphological trait, we would be content with having a single measurement per species! I can't remember if the authors consider variation in coloration among the sexes either...

>>We understand that there are often plumage differences between subspecies and sexes. The nominate subspecies is not always “the most widely distributed”, but in the absence of making a judgement call to decide which subspecies best fits that description, we feel this is a reasonable analytical choice. Note that there are currently no phylogenetic comparative methods that can comprehensively handle different trait values (as opposed to intraspecific variation) within a species in a multivariate distance framework, and that there are currently no subspecific-level woodpecker phylogenies; splitting the tips into polytomies would lead to spurious conclusions, as it would imply fairly notable evolutionary change over zero evolutionary time. After solving these problems, both of these questions would make for fantastic follow-up studies, but they are beyond the scope of the current study. Additionally, that we uncovered a strong signal here despite our conservative approach not being able to adequately incorporate known subspecific cases of mimicry lends support to the strength of our results.

Sexual selection. Many studies have pointed out to a tradeoff between size dimorphism and sexual dichromatism in birds. Furthermore... many of the most sexually selected birds we know are dichromatic but not sexually dimorphic. This manuscript uses only size dimorphism as a metric sexual selection and therefore cannot really make any general statements about the role or lack thereof sexual selection on the processes being studied. At the very least some discussion on the topic would be warranted.

>>We agree with the reviewer and have extensively caveated our language here (lines 535-537).

The paragraph starting in ln 65 talks about effects of total precipitation on how dark species are and how they are patterned. However, the paragraph starting in line 85 talks again about darkness (colorPC1) but mentions that the best model for it included only body mass, latitude, and seasonality in precipitation... what am I missing? Was total precipitation not part of this best model? Why then were two different models used to

analyze this variable. Apologies if I am missing something obvious but the way this is currently written makes it very difficult to understand which results come from different analyses. A similar issue is observed for pattern variables.

>>We thank the reviewer for pointing out this lack of clarity. We've revised these lines (now, lines 80-84 and 103-105) to point the reader to the same figure panels in each case—the significant correlate is the precipitation-related axis from the climate PCA, which has total precipitation strongly loaded on one side, and seasonality in precipitation on the other.

Reviewers' Comments:

Reviewer #1:

Remarks to the Author:

The revised article is now clearer in its scope and generality, and offers general support for known plumage correlations with climate, as well as a number of novel correlations that bear further investigation.

I am still bemused as to the mechanism that leads to the "profound" if highly constrained patterns in Figure 4, but am willing to let the community respond.

Reviewer #3:

Remarks to the Author:

I am Reviewer # 3 in the last round of peer-review. After reading the new version of the manuscript I am happy to see that the authors have paid close attention to reviewer's suggestions and have made reasonable adjustments whenever possible. This project is impressive in many ways and it is nice to see that the various checks that were suggested did not substantially alter the main results or the main message. I must also add that having toned down the claims of novelty and adjusting the title has given the study more weight without sacrificing its appeal.

There is one possible methodological issue that remains and I sincerely apologize for not having raised it in my last round of comments (I am truly sorry... at that time I focused on addressing data-collection aspects of the study, which seemed to be the most pressing matter). All of the major findings in this manuscript are based on variations of the Mantel test. The authors cite Legendre's work (specifically his 2012 paper), who famously concluded that it was OK to use these methods. However, I must point out that there is significant disagreement among ecologists and evolutionary biologists on whether that conclusion is true or at least on the conditions under which it is. For example, Guillot & Rousset (2013) Dismantling the Mantel tests. *Methods in Ecology and Evolution*, 4: 336–344, as well as Harmon & Glor (2010) conclude that Mantel tests and partial Mantel tests suffer of grossly inflated type I error rates, meaning that more often than expected, you find significant associations where there are none. I therefore believe that it is CRITICAL that the authors address this issue head on in their paper with an explicit acknowledgement of the problem as well as with some discussion on why it may or may not apply to their specific analyses, and on what it means if it does. Please note that this is not a damning criticism in my opinion... I understand that there is simply no other currently available method that can deal with distance matrices like the ones this study relies on. The findings are very cool and worthwhile publishing but the caveat I mention is real and should be acknowledge so that this paper better stands the test of time.

Thanks for an interesting read

Reviewer 1 stated: “I am still bemused as to the mechanism that leads to the ‘profound’ if highly constrained patterns in Figure 4, but am willing to let the community respond.”

Reviewer 3 stated: “All of the major findings in this manuscript are based on variations of the Mantel test. The authors cite Legendre's work (specifically his 2012 paper), who famously concluded that it was OK to use these methods. However, I must point out that there is significant disagreement among ecologists and evolutionary biologists on whether that conclusion is true or at least on the conditions under which it is. For example, Guillot & Rousset (2013) Dismantling the Mantel tests. *Methods in Ecology and Evolution*, 4: 336–344, as well as Harmon & Glor (2010) conclude that Mantel tests and partial Mantel tests suffer of grossly inflated type I error rates, meaning that more often than expected, you find significant associations where there are none. I therefore believe that it is **CRITICAL** that the authors address this issue head on in their paper with an explicit acknowledgement of the problem as well as with some discussion on why it may or may not apply to their specific analyses, and on what it means if it does.”

In response to these similar points, we added caveats (with citations) to our Mantel test results stating that they warrant further exploration with alternative methodologies. These caveats can be found on lines 550-553.